# Prompting Strategy Use and Beyond: Examining the Relationships between Elaboration, Quantity, and Diversity of Learning Strategies on Performance

**DOI:** 10.3390/bs14090764

**Published:** 2024-09-02

**Authors:** Makai A. Ruffin, Ryann N. Tudor, Margaret E. Beier

**Affiliations:** Department of Psychological Sciences, Rice University, Houston, TX 77005, USA; rnt2@rice.edu (R.N.T.); beier@rice.edu (M.E.B.)

**Keywords:** elaboration, recall performance, generative learning strategies

## Abstract

Elaboration is a generative learning strategy wherein learners link prior knowledge and experiences with to-be-remembered information. It is positively related to an array of learning outcomes. However, most students do not independently use generative learning strategies. We explored whether prompting elaboration learning strategies when reading an academic passage influenced knowledge test performance. Participants were randomly assigned to two conditions: receiving a prompt (i.e., experimental; *n* = 94) and no prompt (i.e., control; *n* = 112). The results revealed that participants who received the elaboration prompt (*M* = 13.88, *SD* = 2.20) did not outperform learners who did not receive the prompt (*M* = 13.67, *SD* = 2.43) on the knowledge test. However, we did find a positive relationship between the extent of elaboration strategy use and knowledge test performance across conditions (*r* = 0.17, *p* < 0.05). Twelve themes emerged from an exploratory thematic analysis, wherein participants were asked about the learning strategies they used when reading the passage. Students used a variety of learning strategies unprompted, although 42.15% reported not using any additional learning strategies outside of the prompt or using low-utility learning strategies (e.g., relying on memory, skimming). Further exploratory analyses found that the quantity and diversity of learning strategies used individually influenced knowledge test performance. ANCOVA results revealed, however, that when controlling for quantity, the diversity of learning strategies used did not significantly influence knowledge test performance. Our findings contribute to prior literature by (1) demonstrating a relationship between elaboration strategy use and test performance, (2) highlighting learning strategies students use to retain information, and (3) exploring additional factors regarding learning strategy use that influence performance.

## 1. Introduction

Learning is broadly defined as a process that creates and changes one’s knowledge, skills, values, and worldviews [1], and students and instructors are constantly seeking to improve learning outcomes. However, many learners lack knowledge of effective learning strategies and often do not incorporate effective strategies into their learning routines [2]. Given that learners typically are not formally trained on how to use effective strategies [3], the question becomes: Can learners be prompted to use effective learning behaviors or techniques?

The current study explores the impact of prompting generative learning strategies, which involve learners making sense of information to be learned by selecting relevant information, organizing it meaningfully and coherently, and integrating it into existing knowledge [4,5]. In particular, we examined whether prompting the use of generative learning strategies (i.e., overt behavioral activities aimed at supporting understanding during learning [6]) influences performance on a subsequent knowledge test. Additionally, using thematic analysis [7], we explored the learning strategies learners report using in addition to those prompted. Further, we explored the impact of the quantity and diversity of learning strategies used on performance. In doing so, we contribute to the existing literature on generative learning strategies by (1) examining if prompting generative learning strategy use improves performance, (2) exploring the learning behaviors people naturally report, and (3) investigating if the quantity and diversity of learning strategies used impacts performance. In what follows, we provide a conceptual background of generative learning theory to examine how cognitive processes may influence recall, learning, and performance outcomes.

### 1.1. The SOI Model of Generative Learning

One of the core tenets of generative learning theory is that people invest effort to integrate to-be-learned information and experiences into existing knowledge structures [8]. Learners who engage in generative learning actively build relationships between the to-be-learned material and existing knowledge, beliefs, and experiences [8]. According to Wittrock’s model, generative learning consists of four processes: attention, motivation, memory, and generation [9]. Building relationships between to-be-learned information and existing knowledge (i.e., generation) does not take place without directing attention towards generative processes that engage the to-be-learned material, a willingness to invest effort towards making sense of to-be-learned material (i.e., motivation), and the learners’ prior knowledge, experiences, and beliefs [10,11].

Closely tied to Wittrock’s model of generative learning is the Select–Organize–Integrate (SOI) model of generative learning, which describes the processes through which generative learning strategies are effective. The SOI model argues that generative learning strategies engage three sequential cognitive processes: selective attention, organization, and knowledge integration [6]. That is, learners must first select relevant content stored in sensory memory to be used later in working memory, coherently organize it so they can easily process and retain it, and integrate it into existing knowledge structures. Fiorella and Mayer [11] review the effectiveness of generative learning activities (i.e., those involving selecting, organizing, and integrating to-be-learned information). They found that these strategies result in medium to large standardized mean differences between experimental and control groups on various learning outcomes and are particularly effective for undergraduate students [11,12].

The current study focused on the effectiveness of prompting the generative learning strategy of elaboration. Elaboration is a learning strategy wherein learners connect their prior knowledge and experiences with to-be-remembered information [13,14]. Elaboration involves paraphrasing, creating analogies, and generative note-taking [15]. The connections between prior knowledge and to-be-remembered information allow for multiple retrieval routes when recalling information, such that learners can recall forgotten information by reconstructing it from connections previously made [13]. Further, elaboration is argued to allow for deeper processing compared with more passive learning strategies (e.g., highlighting text) because elaboration requires more effort from the learner [16]. In sum, elaboration strategies aim to change how the learning material is viewed and understood in the context of prior knowledge to increase initial levels of understanding [14]. Thus, elaboration strategies involve selecting, organizing, and integrating new experiences with existing knowledge structures—a central tenet of generative learning theory.

### 1.2. Effectiveness of Prompting Elaboration Strategy Use on Learning Outcomes

Prior work has explored the effectiveness of prompting elaboration strategy use on several learning-related outcomes. Prompts are “questions or hints that induce productive learning processes” [17] (p. 566), which tend to be effective because they can help overcome superficial processing and induce specific learning strategy behaviors and approaches [17,18]. Studies have conceptualized and delivered prompts in several ways. For example, in a sample of undergraduate students, Hannon [19] compared the effectiveness of differential-associative processing (i.e., a form of elaboration wherein learners identify differences between definitions or concepts) and example elaboration (i.e., comparing examples of a concept and identifying critical features of the examples). The results of this study suggest that prompting is effective for engaging either learning strategy.

Vogt and colleagues [20] also investigated how prompting elaboration before participating in an immersive virtual reality learning environment influences learning robotics. Participants who received the elaboration prompt were told to pay attention to new concepts, think about how they may be related, and imagine that they would later have to explain the concepts to someone without prior knowledge of robotics. In a sample of undergraduate students, they found that those who received the elaboration prompt had increased recall of simple terms (i.e., knowledge), but not increased performance for higher-level performance, including comprehension and application. Johnsey and colleagues [21] also found in an adult sample that those who were prompted to elaborate on their own had higher recall, recognition, and application of topics on supervisory communication compared to those who were prompted to use elaborations created by the instructor and those who did not receive a prompt. These findings highlight how prompting the use of elaboration is beneficial not just for undergraduate student samples, but also for adult learners. 

Further, Berthold and colleagues [17] investigated the effect of elaboration, organization, and metacognitive prompts on understanding and retention. Using a sample of undergraduate psychology students, participants in the elaboration condition were prompted to (1) create examples that illustrate, confirm, or conflict with the learning content, (2) create links between the learning content and their knowledge from school and everyday experiences, and (3) describe aspects of the learning content that they found interesting, useful, and convincing (as well as aspects that did not have these qualities). They found that participants who received prompts encouraging elaboration and organization strategy use outperformed those who did not receive prompts on tests of understanding and retention. 

In sum, elaboration techniques encourage learners to actively construct connections between new information and existing knowledge and foster a deeper understanding of the material. Our research question is as follows: Do prompts that require learners to connect to-be-learned material with personal experiences influence learning outcomes, particularly performance on a knowledge test? Adapted from the prompt administered in the study by Berthold and colleagues [17], we hypothesized that receiving elaboration strategy prompts that direct learners to connect to-be-learned information to their own personal experiences would promote increased encoding and retrieval of information, leading to enhanced learning outcomes.

**Hypothesis 1**:
*Participants who receive a prompt to create examples that connect to their personal experiences will outperform participants who do not receive a prompt on a knowledge test.*


### 1.3. What Additional Strategies Are Learners Using When Unprompted?

In addition to examining the impact of elaboration strategy prompts on knowledge test performance, we explore strategies participants use outside of the elaboration prompt. Work from Endres and colleagues [22] examined the impact of elaboration prompts on recall and comprehension outcomes. Although participants who received the prompt outperformed participants who did not in comprehension, both groups used additional learning strategies outside of the circumscribed prompt. In this case, participants also used organization (i.e., identifying relationships between main ideas) and metacognitive (i.e., planning, monitoring, and evaluating learning behaviors) strategies during the intervention. Although prompts are intended to induce behaviors that learners do not spontaneously and effectively demonstrate [18], learners may still use additional strategies depending on several factors, such as the amount of time, the required level of mastery, and their belief that learning strategies are effective [23,24]. Thus, we explore what additional strategies learners may use with the following research question: 


**Research Question 1**
*: What strategies are participants using outside the elaboration prompt?*


### 1.4. Effectiveness of Quantity and Diversity of Learning Strategy Use on Learning Outcomes

Furthermore, we explore two aspects of learning strategy use: quantity and diversity. Work from Walck-Shannon and colleagues [25] found that when controlling for course preparation, course absences, and total study time, the number of active learning strategies participants used resulted in increased exam scores. Further, Glogger and colleagues [26] found that the quantity of learning strategies used was predictive of test performance, particularly when those strategies were associated with deeper processing (i.e., elaboration and organization).

Although previous studies have found a significant effect of the quantity of learning strategy use on performance, which supports the argument that the more links learners create between new information and prior knowledge, the higher their learning outcomes, quantity may not be more important than other aspects of learning strategies. For example, in the context of note-taking, Winne and Perry [27] argue that a large quantity of notes may indicate that students are not selective when documenting relevant content. As such, in the following research question, we explore the impact of the quantity of learning strategies used on knowledge test performance.

**Research Question 2**: *Does the quantity of learning strategies used predict knowledge test performance?*

Glogger and colleagues [26] also explored how combinations of learning strategies influence test performance. Using cluster analyses, they found that the most successful students used a combination of cognitive (i.e., elaboration, organization, rehearsal) and metacognitive (i.e., self-monitoring) learning strategies. The most successful students also used more learning strategies. Less successful students in this study used fewer learning strategies or strategies that have been found to be ineffective (e.g., re-reading). Work from Berthold [17] and Nückles and colleagues [28] also found that a combination of learning strategies improved comprehension and retention outcomes. Thus, in addition to quantity, our third research question also explores if the diversity of learning strategies reported influences learning outcomes. 


**Research Question 3**
*: Do participants who use a diversity of learning strategies outperform participants who solely use (1) generative or (2) non-generative learning strategies on a knowledge test?*


## 2. Methods

### 2.1. Participants

Participants enrolled in at least one psychology course had the option to engage in the current study advertised among a broader pool of existing psychological research opportunities. Initially, 278 undergraduate students enrolled at a university in the southern United States participated in the study. Eleven participants were removed due to an error in study administration (i.e., they were not assigned to a condition), and five participants were removed for missing responses on the final knowledge test. Further, 38 participants in the experimental condition and 18 in the control condition failed the attention and manipulation checks and/or had responses outside three standard deviations from the mean on the prior knowledge and final knowledge test variables and were removed from the study. Upon completion of the study, participants were awarded course credit allocated towards fulfilling degree requirements.

Our final sample consisted of *N* = 206 participants (*n* = 94 experimental group and *n* = 112 control). There was no significant difference in gender ($X^{2}$(3) = 1.62, *p* = 0.656), age (*t*(199) = −1.30, *p* = 0.196), education (*t*(191) = 0.01, *p* = 0.995), or race/ethnicity ($X^{2}$(6) = 2.87, *p* = 0.826) between the experimental and control groups. Further, we found no difference in prior knowledge (*t*(4.19) = 0.55, *p* = 0.612) between participants with complete data (*n* = 206) compared with those who only completed the prior knowledge assessment (*n* = 5). On average, our final sample was approximately 19 years old (*M* = 19.36, *SD* = 1.15); 63% identified as female, 97% earned a high school diploma, and 44% identified as Asian.

### 2.2. Materials

OpenStax provides free, open education resources (e.g., textbooks [29]) that span a wide range of subjects such as computer science, nursing, mathematics (e.g., calculus, statistics), humanities (e.g., philosophy, world and U.S. history), science (e.g., physics, biology), and social sciences (e.g., psychology, economics). We selected a passage related to Industrial-Organizational (I-O) psychology for the current study. I-O psychology is the scientific study of human behavior in organizations that addresses issues in the workplace, such as recruitment, selection, training and development, performance management, well-being, leadership, and work motivation [30]. I-O psychology is also a fast-growing segment in psychology—the Occupational Outlook Handbook issued by the Bureau of Labor Statistics predicts that employment in I-O psychology-related occupations will increase from 2022 to 2032 by approximately 6% [31]. Because it is relevant to the workplace many students aspire to, we thought I-O psychology would be broadly interesting to the undergraduates participating in this study.

All participants read a 1583-word passage about I-O psychology that is part of the OpenStax psychology open-access textbook [29]. The passage topics were employee selection, candidate testing, and workplace training. We selected this passage because these topics were relatable to students’ personal experiences and prior knowledge and would, thus, be appropriate for the elaboration intervention.

### 2.3. Procedure

This study was conducted online via the Qualtrics platform between November 2022 and February 2023 (see Figure 1 for our research design). First, all participants were asked to report their prior knowledge of concepts relevant to the field of I-O psychology. Next, participants were randomly assigned to either the experimental or control condition. Participants in the experimental condition were prompted to use an elaboration learning strategy we adapted from Berthold and colleagues [17], which stated, “You are now going to read a brief section from an introductory psychology textbook. Please grab a scrap piece of paper. On this paper, create and write down personal examples from your life that illustrate, confirm, or conflict with the information presented in the passage. You will use your examples from this learning strategy to help you recall the information from the passage later on in the study.” Participants in the control condition did not receive this prompt. Instead, they were instructed to spend their time reading the passage because they would be asked to recall the information presented later in the study. After the instructions, participants in both conditions were given 8 min to read the passage with I-O psychology-related concepts as many times as preferred, but were not given the option to move forward until the eight-minute timer expired. The experiment took approximately 30 min to complete.

After reading the passage, both conditions were given a five-minute waiting period. During this period, all participants engaged in a distractor tile game task called 2048 [32], wherein they strategically slid identical tiles together to create a larger number (e.g., putting two four tiles together to create an eight tile) to reach a sum of 2048. After this waiting period, participants in both conditions completed the final knowledge test, recalling details presented in the I-O passage read during the experiment. Both conditions were then given attention and manipulation checks, and they were asked to report the extent to which they created personal examples that illustrated, confirmed, or conflicted with the information presented in the passage. We also included an open-ended question that asked participants in both conditions to report any additional learning strategies they used to remember the information presented—which we analyzed in an exploratory fashion. 

### 2.4. Measures

(a) Prior I-O Psychology Knowledge: To measure prior knowledge of I-O psychology concepts, we used content discussed in the passage and existing test bank questions from the OpenStax psychology textbook [29]. The original list consisted of 20 items; however, after discussions with subject matter experts (i.e., faculty and graduate students with expertise in I-O psychology), 16 concepts were retained, including job analysis, candidate selection methods, and training effectiveness. Participants were asked to rate their knowledge of concepts related to I-O psychology knowledge on a 5-point Likert scale (e.g., 1 = *I know very little* to 5 = *I know a great deal*; α = 0.96). Scores were calculated by taking the average rating across all 16 items for each participant. The minimum score in our sample was 1.00, and the maximum score = 4.00.

(b) Final Knowledge Test Performance: Similar to the measure of prior I-O psychology knowledge, we drew from the existing test bank from the OpenStax Psychology textbook and content from the passage to create multiple-choice items to test recall of the passage content. The initial item pool consisted of 24 items, and as a result of subject matter expert input, we dropped one item, resulting in 23 multiple-choice questions. An example item is, “Which of the following is included in the O*NET?” Answer options were: (A) Occupation tasks, (B) Education requirements, (C) Work context, and (D) All of the above (with D being the correct item response).

Internal reliability was estimated using the Kuder–Richardson formula-20 (KR20), which is most appropriate for dichotomously scored (i.e., incorrect/correct) items [33]. The KR20 for the 23-item measure was 0.49, which was below typical conventions (i.e., 0.70 threshold). To assess the low KR-20 estimate, a distractor analysis was conducted to evaluate the performance of each multiple-choice answer option [34] using the distractor analysis functionality in the *snowirt* package in Jamovi [35]. The results suggested that five items had a negative item-total and rest correlation and were removed from the measure. After removing these items, the KR20 was 0.60, which is still relatively below standard conventions.

The final knowledge test performance was scored with the 18 items remaining, and participants received one point for each correct response. Thus, the maximum score participants could receive was 18 (meaning they answered every multiple-choice question correctly). The maximum score achieved across conditions was 18, and the minimum was 7. In the discussion section, we revisit possible limitations of low reliability estimates for our outcome of interest. 

(c) Extent of Elaboration Strategy Use: Participants were asked, “To what extent when reading the passage did you create personal examples from your life that illustrated, confirmed, or conflicted with the information in the passage?” Responses ranged on a 5-point Likert scale (1 = *None at all* to 5 = *A great deal*). The minimum score in our sample was 1.00, and the maximum score = 5.00. 

(d) Additional Self-Reported Learning Strategy Use: Participants were asked, “What learning strategies did you use (if any at all) to remember the information presented in the passage?” Participants were shown a text box wherein they could report additional learning strategies they may have used to retain the information presented in the passage.

(e) Attention and Manipulation Check: The attention check was a single-item measure wherein participants were asked to select “strongly agree” if they were paying attention to the question. We wanted to ensure participants paid attention to the prompt (or lack thereof) for the manipulation check. All study participants were administered a multiple-choice question that asked, “What learning strategy were you instructed to use?” Participants in the experimental condition had to select the option, “Creating examples that illustrate, confirm, or conflict with the information presented in the passage,” and the control condition had to select the option, “I was not instructed to use a learning strategy when reading the passage.” Incorrect responses to these answers resulted in participants being removed from further analyses.

### 2.5. Data Analysis

All analyses were conducted in R Studio (Version 2023.09.1). To test Hypothesis 1, we conducted an independent samples *t*-test to examine whether there was a statistically significant difference in average knowledge test performance between the experimental and control groups. We also conducted an independent samples *t*-test in our first exploratory analysis, wherein we explored group mean differences in the extent of elaboration strategy use. In the second exploratory analysis, we conducted a Pearson correlation to determine the strength and direction of the relationship between the extent of elaboration strategy use and knowledge test performance. 

To explore the additional learning strategies participants reported when reading the passage, we adopted an inductive thematic analysis approach to analyze our open-ended responses. This analysis allows researchers to analyze and identify themes derived from the open-ended response data without any expected or preconceived notions about the outcome [7]. The inductive thematic analysis process involves five steps: (1) familiarizing oneself with the data, (2) generating initial codes, (3) searching for themes that stand out, (4) reviewing themes that were generated, and (5) defining and labeling the themes that are critical and relevant to understanding learning strategy use [7]. To demonstrate the consistency of the application of codes across coders, the first author went through the open-ended responses to familiarize themselves with the data. Then, the same researcher generated first-order codes (i.e., labels with descriptive information that is most often a word or phrase [36]). Then, categories were established based on the properties of the codes that were both similar and distinct, and a coding dictionary was developed that arranged the first-order codes into aggregated second-order codes (i.e., categories that explain the pattern of the first-order codes [37]). We then provided four undergraduate student coders, unfamiliar with the study objectives, the coding dictionary and participant open-ended responses. We asked the coders to assign a code to each open-ended response. The overall agreement among the coders was 0.74—slightly above the minimum requirement of 0.70 [38]. Discrepancies were discussed among the first and second authors of the paper, and a final code was assigned. Finally, the second-order codes were placed into broader themes representing the dataset’s patterned responses [7].

To examine the relationship between the quantity of learning strategies reported and knowledge test performance, the number of learning strategies was counted based on participants’ open-ended responses, regardless of whether they were generative. Participants who only reported antecedents of generative learning were coded as zero additional learning strategies reported because they were considered metacognitive and motivational processes that initiate, monitor, and direct cognitive resources when learning [11], rather than engaging in behaviors that allow deeper processing and application to new situations [39].

To explore the relationship between the diversity of learning strategies used and knowledge test performance, participants’ open-ended responses and themes from the thematic analysis were used to place participants into one of three groups: solely reporting the use of non-generative learning strategies (e.g., repetition), (2) solely reporting the use of generative learning strategies (e.g., summarization), or (3) reporting a mix of non-generative and generative learning strategies (e.g., repetition and summarization). Because the goal was to compare average knowledge test performance among the three groups, an ANOVA was conducted. Finally, we ran an ANCOVA to control for the effect of the quantity of learning strategies used on knowledge test performance.

## 3. Results

Means, standard deviations, effect sizes (i.e., Cohen’s *d*’s), and scale ranges of study variables by condition are in Table 1. Beginning with prior knowledge of I-O psychology concepts, there was not a significant difference (*t*(204) = −1.54, *p* = 0.126) between those who were prompted to use elaboration strategies (*M* = 1.53, *SD* = 0.64) compared with those who were not (*M* = 1.68, *SD* = 0.73). This suggests that both conditions were similar in their prior knowledge of the field of I-O psychology before engaging in the experiment, which was expected and preferred, given that we randomly assigned participants to each condition.

Contrary to expectations, there was not a statistically significant difference (*t*(203) = 0.66, *p* = 0.509, *d* = 0.09) between the experimental condition comprised of those instructed to use elaboration strategies (*M* = 13.88, *SD* = 2.20) and the control condition (*M* = 13.67, *SD* = 2.43). This result contradicts our hypothesis that the elaboration strategy condition would outperform the control condition on the knowledge test.

### 3.1. Exploratory Analyses 

#### 3.1.1. Extent of Elaboration Strategy Use

We also analyzed the extent of elaboration strategy use in both conditions. We found a statistically significant difference (*t*(204) = 2.87, *p* = 0.005, *d* = 0.40) between those who were instructed to use elaboration strategies (*M* = 2.45, *SD* = 0.89) compared with those who were not (*M* = 2.05, *SD* = 1.08; see Figure 2). Overall, however, the mean values indicate that participants reported relatively low rates of elaboration strategy use regardless of condition.

#### 3.1.2. Relationship between Extent of Elaboration Strategy Use and Knowledge Test Performance

Although we did not find evidence for a causal relationship between prompting elaboration strategies and knowledge test performance, we were interested in exploring whether the study variables were related at the zero-order (see Table 2). Prior I-O knowledge was not significantly related to knowledge test performance (*r* = 0.02, *p* = 0.781). Surprisingly, the extent of elaboration learning strategy use and final knowledge test performance were positively related (*r* = 0.17, *p* < 0.05). That is, when participants reported higher usage of the elaboration strategy, their knowledge test performance was more likely to increase regardless of condition. To understand the relationship between the extent of elaboration learning strategy use and performance on the knowledge test independent of prior knowledge, we also examined partial correlations among study variables. The patterns of relationships did not differ, however, and are not reported here.

#### 3.1.3. What Learning Strategies Do Learners Report Using When Reading Expository Text?

We were interested in the learning strategies students used unprompted when reading and retaining the passage information. To this end, twelve themes (reflection, heightened focus, self-regulation, environment management, organization, summarization, connection, animation, localization, documentation, no or low-utility strategies, and repetition) were organized into three broad categories.

Antecedents of generative learning strategy use;Reported strategies aligned with generative learning strategies, per the SOI model;Ineffective learning strategy use.

##### Antecedents of Generative Learning Strategy Use

According to the SOI model, metacognition and motivational processes are the sources that enable learners to initiate, monitor, and direct cognitive resources when learning [11]. When learners are aware and motivated to select, organize, and integrate to-be-learned material into existing knowledge structures, they are better able to engage in the generative learning strategies intended to improve learning and performance outcomes. Further, prior research has found that when learners intentionally plan their learning goals and monitor their current knowledge and understanding, they are more likely to use learning strategies [40]. Four themes, which accounted for approximately 14.21% of all coded responses, had metacognitive and motivational components: reflection, heightened focus, self-regulation, and environment management. 

(a) Reflection (approximately 1.47% of all responses coded) is a component of metacognition [41]. Two second-order codes fell under the broader theme of reflection: repeatedly checking understanding and self-questioning. Both checking understanding and self-questioning required the learner to monitor and evaluate what they had retained when reading the passage. For instance, one participant stated, “[I asked] myself questions about the passage as I read” (P3272).

(b) Heightened focus (approximately 8.82% of all coded responses) includes techniques that hone attentional resources to the task. Three second-order codes were included: remembering the first and last pieces of information, focusing on keywords, and focusing on interesting or surprising information in the passage. In reference to focusing on keywords in the passage, one participant said, “I just focused on keywords and phrases that seemed important or repetitive” (P2928).

(c) Self-regulation (approximately 3.43% of all coded responses) is the “modulation of affective, cognitive, and behavioral processes throughout a learning experience to reach a desired level of achievement” [42] (p. 421). Two second-order codes were included: self-motivation and regulating reading pace. For self-motivation, participants attempted to make themselves more interested in the passage information, with some sharing that their interest did increase after reading the passage. Participants also attempted to regulate their reading pace, stating, “I read at a slower pace than I probably could have” (P3244).

(d) Environment management (approximately 0.49% of all coded responses) is when learners monitor and create a distraction-free learning environment [36,37]. It comprised one second-order code: minimizing outside distractions. Participant 2968 said, “I minimized outside distractions and tried to interpret what I was reading while I was reading it instead of simply reading words mindlessly.”

##### Generative Learning Strategy Use

The following six themes, which accounted for approximately 24.01% of all coded responses, provided evidence that participants naturally engaged in learning behaviors that connected with the SOI model of generative learning: organization, summarization, connection, animation, localization, and documentation.

(a) Organization consisted of the learner making meaningful connections between course concepts and structuring the passage content. Five second-order codes—approximately 5.39% of all coded responses—were included: relying on passage headings, blocking out sections of material, creating categories out of information in the passage, grouping information together, and outlining. In reference to relying on passage headings, one participant said, “The passage was organized by headings and so I sort of remembered where each section was about and where it was in relation to other sections in the passage and then when a question had me recall information I looked to see where it was in the spatial organization of my mind” (P1614).

(b) Summarization included capturing the main points throughout the passage and creating brief restatements of passage content in their own words. Two second-order codes (approximately 4.41% of all responses) were included: summarizing information and finding the main points in the passage. One participant said, “[I make] myself summarize and word what I had read in my own terms” (P3344).

(c) Connection was approximately 4.90% of all coded responses. Three second-order codes were included: using prior knowledge and real-world examples, relating to stories and memories, and degree of agreement and disagreement with passage information. Although similar to elaboration, connection included strategies that linked prior experiences, knowledge, beliefs, and attitudes to information presented in the passage. Regarding relations to prior knowledge and real-world examples, one participant said, “[I just related] to previous knowledge about job interviews and training? If there was a newer concept, I felt less likely to answer those accurately” (P2883).

(d) Animation (approximately 2.94% of all coded responses) consisted of participants making the passage content “come alive” by manipulating the information to have memorable patterns, visualizations, or humorous representations. Three second-order codes were included: mnemonics, drawing pictures related to information in the passage, and making jokes to remember the information. Most notably, one participant said, “I made jokes and related stories/memories to pieces of important information from the passage in my notes to help me recall the info” (P3202).

(e) Localization was defined by only one second-order code: visualizing the location of information in the passage, which involved imagining the physical location of where the information was stated. Approximately 3.92% of all coded responses were learners engaging in localization. Specifically, one participant said, “I tried to remember the general area of the article that discussed the topic asked in the question and then attempted to answer the question from there” (P2877).

(f) Documentation included taking notes solely about the passage information. It emerged as an important theme for how individuals recalled information from the passage. Three second-order codes (accounting for approximately 2.45% of all coded responses) were placed under the broader documentation theme: note-taking, annotating the passage, and creating bullet points. Although each second-order code seemed similar, there were small differences in how participants used these strategies in relation to the passage. Each requires the learner to record information on a separate sheet of paper or directly onto the passage’s text. When asked what additional learning strategies were used when reading the passage, one participant noted that they “tried to make bullet points in [their] mind” (P2864).

##### Ineffective Learning Strategy Use

We conceptualized ineffective learning strategy use as strategies that either (1) do not require the learner to select, organize, or integrate to-be-learned information as outlined by the SOI model of generative learning or (2) have been considered ineffective by other learning strategy taxonomies [16]. The following two themes were placed under the broader category of lack of effective strategy use: no or low-utility strategies and repetition, which accounted for approximately 61.75% of all coded responses.

(a) No or Low-Utility Strategies consisted of participants not using any additional learning strategies or using methods that have empirically shown low utility (i.e., they are inconsistent in increasing learner performance [16]). Three second-order codes were included: no additional learning strategies were used, relied on memory, and skimming, which accounted for 42.15% of coded responses. For instance, one participant said, “I didn’t use any, I just tried to thoroughly read the passage” (P1471).

(b) Repetition (also referred to in the literature as rehearsal strategies) includes the mental reiteration and copying of information [14]. Four second-order themes were included, accounting for approximately 19.60% of all coded responses: repeating information, rereading the passage, reading keywords out loud, and attempting to memorize the material. One participant said that they “repeated key terms and definitions [in their head]” (P990). Although repetition may be useful for simple tasks (e.g., rote memorization), it is not as effective for complex tasks such as recalling information presented in academic text [16].

#### 3.1.4. Does the Quantity of Learning Strategies Used Influence Performance?

In addition to categorizing the additional learning strategies reported by the participants described above, we explored whether the quantity of learning strategies reported predicted final knowledge test performance. Participants who only reported using antecedents of generative learning strategy use were removed from exploratory analyses in Section 3.1.4 and Section 3.1.5. We define the quantity of learning strategies used as the number of strategies reported in the participants’ open-ended responses. The minimum number of learning strategies reported was 0, and the maximum number of learning strategies reported was 5 (*M* = 0.72, *SD* = 0.87). Across conditions, we found a positive relationship between the quantity of learning strategies used and final knowledge test performance (*r* = 0.22, *p* < 0.01). Further, the quantity of learning strategies used predicted knowledge test performance (β = 0.61, *p* < 0.01).

#### 3.1.5. Does a Mix of Generative and Non-Generative Learning Strategies Improve Performance?

To understand the efficacy of engaging in multiple learning strategies, we explored whether a mix of generative and non-generative learning strategies led to significant differences in performance. Participants belonged to one of three groups: (1) solely reporting the use of non-generative learning strategies (*n* = 113), (2) solely reporting the use of generative learning strategies (*n* = 32), or (3) reporting a mix of non-generative and generative learning strategies (*n* = 16). 

There was a significant effect of group on final knowledge test performance at the *p* < 0.01 level for the three learning strategy groups (F(2,158) = 5.86, *p* = 0.004). Post-hoc comparisons using the Tukey HSD test indicated that participants who used a mix of generative and non-generative learning strategies (*M* = 15.56, *SD* = 1.36) outperformed participants who solely used generative learning strategies (*M* = 13.53, *SD* = 2.11, *p* = 0.014, *d* = 1.09) and participants who solely used non-generative learning strategies (*M* = 13.45, *SD* = 2.48, *p* = 0.002, *d* = 0.90) on the final knowledge test (see Figure 3).

We further examined whether participants classified in the mixed group were likely to use more learning strategies than those in the other groups, and indeed they were: *M* = 0.42, *SD* = 0.56 for the non-generative group; *M* = 1.16, *SD* = 0.37 for the generative group; and *M* = 2.56, *SD* = 0.96 for the mixed group (*F*(2,158) = 102.3, *p* < 0.001). Because the quantity of learning strategies used is confounded with the diversity of learning strategies used, we re-ran the ANOVA examining the effect of learning strategy type controlling for the quantity of learning strategies. The results of this ANCOVA were non-significant (F(2,157) = 1.22, *p* = 0.30, $\eta^{2}$ = 0.01), suggesting that it is the quantity of learning strategies that drives this effect in our sample.

## 4. Discussion

We examined if prompting generative learning strategies, particularly elaboration, influenced performance on a knowledge-based assessment. Grounded in the SOI model of generative learning, we expected participants who were prompted to use elaboration strategies to outperform participants who were not instructed to use any specific cognitive learning strategies. We did not find support for our hypothesis. However, in a series of exploratory analyses, we found evidence that (1) the extent of elaboration strategy use and performance were positively correlated, and (2) the experimental condition reported higher usage of elaboration strategies than the control condition. An exploratory thematic analysis also found that participants used a variety of learning strategies that were either antecedents of generative learning strategy use, generative learning strategies according to the SOI model, or ineffective learning strategies. Using our themes, we found that the number of learning strategies reported (regardless of whether they were generative) was significantly related to final knowledge test performance. Moreover, participants who reported using a mix of generative and non-generative strategies outperformed participants who solely used generative and non-generative learning strategies on the final knowledge assessment, but this effect may be because participants who use a mix of generative and non-generative strategies are also likely to use more learning strategies.

The current study adds to the existing literature on elaboration strategy use and performance and highlights how learners may be underutilizing generative learning strategies or not using them at all. Our findings also demonstrate that the effectiveness of prompting generative learning strategies is not clearly apparent and that examining the quantity and diversity of learning strategies is a potentially fruitful area for further research.

### 4.1. Theoretical and Practical Implications

Our study makes three key contributions to the literature on the impact of generative learning strategies on learning outcomes. First, we found a small, positive correlation between the extent of elaboration learning strategy use and knowledge test performance. It is likely that learners use ineffective learning strategies because they (1) are not formally trained in how to use effective learning strategies and (2) have used learning strategies that led them to believe that ineffective techniques are effective due to metacognitive illusions (i.e., the tendency to believe that easier processing equates to better processing [23]). Although we did not find causal evidence that prompting generative learning strategy use improved performance, the significant correlation between elaboration strategy use and performance has implications for how people are trained to learn effectively.

Second, our thematic analysis revealed that learners use myriad strategies to recall information and that these strategies vary in effectiveness. Notably, participants most often reported using ineffective strategies with historically low utility, meaning they are inconsistent in increasing learner performance [16], or no additional learning strategies. Prior work has also found similar patterns. For example, Karpicke and colleagues [43] asked college-aged students to report their learning strategies when reading a textbook chapter. They found that participants reported re-reading their notes or content from the textbook (an ineffective strategy [16]). Similar to our study, Karpicke et al. [43] found that most participants failed to mention the empirically supported learning strategies (e.g., self-testing) that were previously provided.

Third, although our work demonstrated that many participants did not use additional learning strategies or used strategies with low utility, participants who used a combination of non-generative and generative learning strategies outperformed participants who reported only using non-generative or generative learning strategies. Our findings relative to the mixed learning strategies used may spur interesting research on what combination of generative and non-generative learning strategies (e.g., repetition and summarization versus skimming and organization) yields the highest performance outcomes. Further, our findings support the idea that low-utility learning strategies may alleviate declines in performance when used in tandem with empirically supported generative learning strategies. Because the quantity and diversity of learning strategy use were confounded in our exploratory analysis (i.e., those who used a mix of learning strategies also tended to use more learning strategies), it will be important for future research to tease apart quantity and diversity in an experimental context.

Our work can inform the implementation of generative learning strategies into digital textbook platforms (e.g., OpenStax). Implementation of such strategies requires intensive efforts from online learning platform designers as well as the deliberate pairing of prompts with content provided in online textbooks. Further, digital textbook platforms have offered little to no support for using generative learning strategies [44]. Thus, even if online textbook platforms offer quizzes and other opportunities to learn, there may not be a focus on generative learning strategies, let alone a way to assess the learners and instructional environments that make those strategies most effective. Thus, our findings can inform the design of personalized learning interventions.

### 4.2. Limitations

Our study is subject to limitations. One limitation is related to the prompt itself. The passage used in the experiment pulled from core topics in the field of I-O psychology, specifically the process of hiring, interviewing, and workplace training. Because I-O psychology focuses on applying knowledge from individual, group, and organizational behavior to generate solutions to problems in the workplace [30], it could be argued that there is a misalignment between participants’ ability to meaningfully connect with the passage and the types of experiences that we are asking them to call upon with this particular elaboration strategy prompt. Participants were undergraduate psychology students, so it may be the case that they did not have personal experience with hiring, interviewing, or training practices in a formalized setting, thus making it difficult to fully engage with the prompt’s instructions. Nonetheless, we assume that most participants in the current study aspire to work one day and, thus, may have some interest in the content.

Second, we did not ask participants to report which degree they were pursuing. As a result, it is possible that the participants were psychology majors and/or had prior knowledge of I-O psychology, influencing their knowledge test performance. Nonetheless, there was variability in knowledge test performance and no evidence that performance was at ceiling. If there were effects of prior knowledge, they did not result in different relationships among variables and were likely evenly distributed among conditions, which were randomly assigned.

Third, the reliability estimate of knowledge test performance (i.e., KR20 = 0.60) was below a normative threshold. This may, in part, be due to the nature of the items on the I-O psychology knowledge assessment, in that they had a high degree of heterogeneity. For example, several items on the knowledge test asked participants to recall information about job analyses (e.g., methods for data collection, databases that maintain information about occupations), while other items were focused on workplace training (e.g., forms of training instruction, measurement of training effectiveness). Thus, low internal consistency reliability estimates would be expected when the test content is heterogeneous [45].

Fourth, we did not conduct an a priori power analysis. However, we estimated that we had enough participants to detect a medium effect (Cohen’s *d* = 0.50, power: 1 − β = 0.95), given that we had approximately 100 subjects per condition. We collected data for four months (i.e., November 2022 until February 2023), we were limited to a specific pool of participants enrolled in at least one psychology course, and we attempted to recruit as many participants as possible. Although post hoc power analyses are controversial, given that they are entirely determined by *p*-values, such that high *p*-values tend to have low power (and subsequently, low *p*-values tend to have high power [46,47]), we conducted one because some have argued they are valuable [48]. Given our small effect (*d* = 0.09, *p* = 0.509), the power to find a significant effect was 0.10. One reason our effect was so small may be that subjects may not have been motivated. Thus, we recommend that future researchers examining the effects of prompts consider the sample size implications and gauge participants’ interest. Fifth, our intervention was relatively brief, with one treatment and no retention interval. Specifically, participants who received the prompt only engaged with it for eight minutes. Because of these factors, we cannot speak to important factors such as the sustainability and transferability of elaboration strategy prompts.

Our final limitation is more general. There are inherent issues with designing and conducting experimental research in online environments (e.g., surrounding distractions, internet speed, and different device interfaces [49]). Further, we asked participants to provide their own piece of paper to write down personal examples from their lives that illustrate, confirm, or conflict with the information presented in the passage. Thus, due to the lack of researcher supervision, we cannot verify that the participants adhered to these instructions. However, students are increasingly turning to online learning environments, so it is important to continue examining their methods to gain knowledge and skills.

### 4.3. Future Research Directions

A feature of our study design was having participants engage in a 5 min distraction task before having them take a knowledge test that covered the material from the passage read previously. We did not find that elaboration strategy use influenced knowledge test performance, and perhaps one reason behind this finding is the length of time between the prompt and the measurement of our outcome. Future research should vary the time between the elaboration strategy prompt and the measurement of the knowledge test performance (e.g., one day after the prompt or one week after the prompt). This would allow researchers to understand how elaboration strategies enhance memory recall capabilities and subsequent performance over time. Additionally, although the experimental condition reported a higher extent of elaboration strategy use, this was measured immediately after the intervention. Future research should explore the sustainability of the extent of elaboration strategy use after being prompted. Another reason we did not find an effect of elaboration prompts on performance may be that the learners required more instruction to successfully implement the prompts [16]. This is especially important for learners with low prior knowledge of the content domain because the effectiveness of learning strategies depends on the quality of the elaborations generated by the learner [11]. Future research could explore how varying levels of instruction and subsequent skill at elaboration influence performance.

Another possible future research question is whether the depth of the elaboration used correlates with how well the participant remembers the information from the passage. Further, future research should investigate if familiarity with the passage affects the ability to utilize elaborative learning strategies, which could affect learning outcomes. In doing so, future research could address the effectiveness of elaboration learning strategy use for learners who have little to no prior experience to pull from when using this type of generative learning strategy. Finally, our study did not collect our participants’ notes. We encourage future research to explore what type of experiences participants draw from and write down in their notes when using a similar elaboration prompt to explore what experiences best predict knowledge test performance.

## 5. Conclusions

Many instructors and students focus on the content to be learned and how the students engage with that content [50]. Although we did not find evidence for the effects of prompts, the current study suggests that using elaboration learning strategies can lead to better learning, but that learners may not necessarily know which learning strategies are effective or how to engage in them. Our study underscores the need for further research to explore the practical contribution of prompting elaboration strategy use over time, across different learning contexts, and in tandem with other learning strategies.

## Figures and Tables

**Figure 1 behavsci-14-00764-f001:**
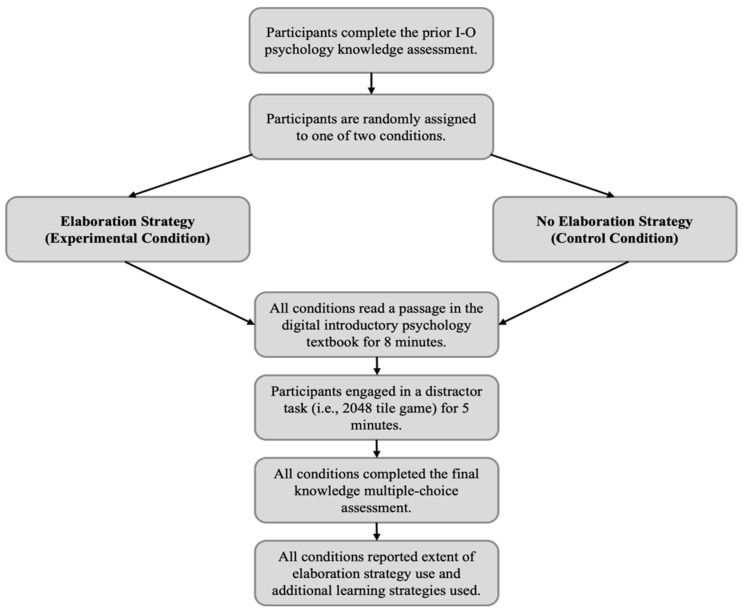
Research design.

**Figure 2 behavsci-14-00764-f002:**
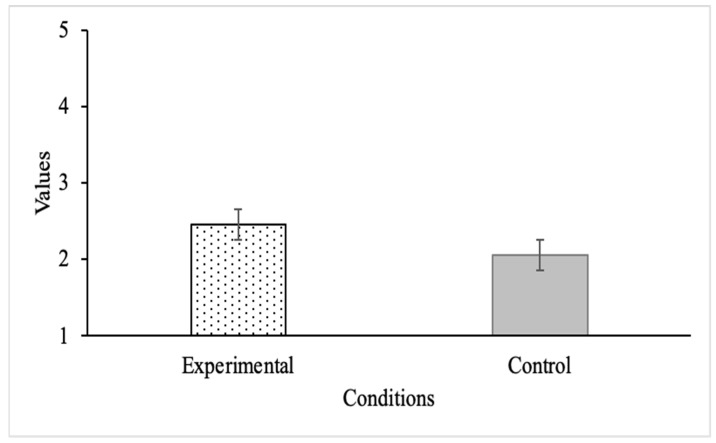
Elaboration strategy use by condition; *n* = 94 for the experimental condition, and *n* = 112 for the control condition.

**Figure 3 behavsci-14-00764-f003:**
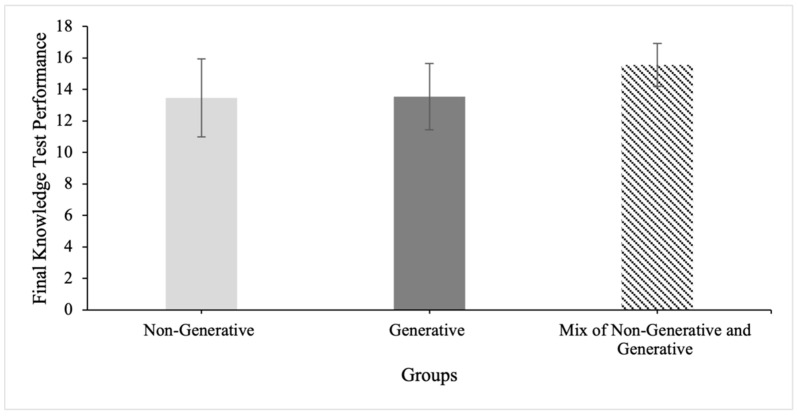
Comparison of participants who used non-generative, generative, and a mix of learning strategies on final knowledge test performance. Error bars represent standard deviations.

**Table 1 behavsci-14-00764-t001:** Means, standard deviations, and Cohen’s *d* of study variables by condition.

	Scale Range	Elaboration Prompt	No Elaboration Prompt	Cohen’s *d*	*p* Value
Variable					
Prior I-O Psychology Knowledge	Likert Scale (1–5)	1.53 (0.64)	1.68 (0.73)	0.21	0.126
Knowledge Test Performance	Multiple Choice (0–18)	13.88 (2.20)	13.67 (2.43)	0.09	0.509
Extent of Elaboration Learning Strategy Use	Likert Scale (1–5)	2.45 (0.89)	2.05 (1.08)	0.40	0.005 **

Note. *n* = 94 for the experimental condition. *n* = 112 for the control condition. Standard deviations are in parentheses. ** *p* < 0.01.

**Table 2 behavsci-14-00764-t002:** Reliability estimates and intercorrelations of study variables.

Variable	1.	2.	3.
1. Prior I-O Psychology Knowledge	(0.96)		
2. Knowledge Test Performance	0.02 [−0.12, 0.16]	(0.60)	
3. Extent of Elaboration Learning Strategy Use	0.11 [−0.02, 0.25]	0.17 * [0.04, 0.30]	-

*Note. N* = 206. * *p* < 0.05. Brackets underneath correlations represent 95% confidence intervals. Reliability estimates are on the diagonal. Cronbach’s alpha was calculated to estimate the reliability of the prior knowledge of I-O psychology measure. KR20 was used to estimate reliability for the knowledge test. Elaboration learning strategy use was a single-item measure.

## Data Availability

The original data presented in the study are openly available in OSF at https://osf.io/yhz9q/?view_only=5bf92cab170b4ff4908c9ee4a6d2ba3c.

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
