# Peer review of "Prompting Strategy Use and Beyond: Examining the Relationships between Elaboration, Quantity, and Diversity of Learning Strategies on Performance"

_behavsci, 2024, doi:10.3390/bs14090764_

Round 1
Reviewer 1 Report (Previous Reviewer 3)
Comments and Suggestions for Authors
The article is interesting, taking into account the suggestions made in a previous review.
Author Response
Comment 1: The article is interesting, taking into account the suggestions made in a previous review.
Response 1: We appreciate you for taking the time to read our work. The suggestions made in previous rounds of review helped us strengthen the quality of our paper while also spurring interesting future research directions. We are glad you agree.
Reviewer 2 Report (Previous Reviewer 2)
Comments and Suggestions for Authors
Thanks for the opportunity to read this paper. It reported how the prompt of using the elaboration strategies related to performance. It also explored the additional strategies students used, and their relationship with test performance. Below are some suggestions:
1. The study consists of three parts. (1) Demonstrating the relationship between elaboration strategy use and test performance; (2) Exploring additional strategies students use; and (3) Exploring the influence of number and diversity of strategies use on test performance. However, the title of the study only covers the first part, and it would be better for authors to revise the title.
2. Like my concern about the title, in the Introduction section, the authors only reviewed the literature related to elaboration strategy use, and only proposed a single hypothesis related to elaboration strategy. I think it is necessary to elaborate in more detail about how the number and the mixed use of learning strategies impact test performance. The research questions and hypotheses corresponding to the remaining research content should also be added.
3. Do all the previous studies reveal the effectiveness of elaboration strategy? Are there any studies that have not found this strategy to improve test performance? And why? I believe they may be useful to provide an explanation for the results of this study.
4. The section of Data Analysis is in lack. The authors should elaborate on what statistical methods and software they used to analyze the data.
5. I am confused about the operational definition and the measurement of the Number of Learning Strategies Used. In my view, the number of strategies refers to the frequency of strategies use. Maybe it would be more appropriate to use the Variability of Learning Strategies Used.
6. In 3.1.5, I am concerned that the factor of Variability of Learning Strategies would confound the results when comparing test performance among the Mixed Group, Generative Group, and Non-Generative Group, as the Mixed Group tended to use more types of learning strategies. An ANOVA test should be conducted after controlling the factors of the Variability of Learning Strategies.
Author Response
Thanks for the opportunity to read this paper. It reported how the prompt of using the elaboration strategies related to performance. It also explored the additional strategies students used, and their relationship with test performance. Below are some suggestions:
We appreciate you taking the time to review and provide suggestions for improving the current state of our work. We have incorporated most of your feedback into the current revision, and believe that the paper has improved significantly based on your feedback. Please see the point-by-point response to your suggestions below. Our response is in bolded font.
Comment 1: The study consists of three parts. (1) Demonstrating the relationship between elaboration strategy use and test performance; (2) Exploring additional strategies students use; and (3) Exploring the influence of number and diversity of strategies use on test performance. However, the title of the study only covers the first part, and it would be better for authors to revise the title.
Response 1: We have changed the title based on your suggestion from “Does Prompting Elaboration Strategy Use Improve Knowledge Test Performance” to “Prompting Strategy Use and Beyond: Examining the Relationships Between Elaboration, Quantity, and Diversity of Learning Strategies on Performance” (see lines 2-4).
Comment 2: Like my concern about the title, in the Introduction section, the authors only reviewed the literature related to elaboration strategy use, and only proposed a single hypothesis related to elaboration strategy. I think it is necessary to elaborate in more detail about how the number and the mixed use of learning strategies impact test performance. The research questions and hypotheses corresponding to the remaining research content should also be added.
Response 2: We appreciate the push to elaborate and justify the need for our exploratory analyses involving the quantity and diversity of learning strategies used. We incorporated your suggestion into the revised manuscript and included two new sections (Sections 1.3 and 1.4).
In Section 1.3 titled “What Additional Strategies are Learners Using Unprompted?”, we discuss prior work that found that learners will use additional learning strategies (i.e., organization and metacognition) in addition to the elaboration prompt they received. Further, we explain why prompts may be ineffective, such that learners may use additional strategies in tandem with the prompt based on various factors (e.g., amount of time remaining, required level of mastery, belief that ineffective strategies are effective). This addition can be found on lines 137-153. In Section 1.4 titled “Effectiveness of Quantity and Diversity of Learning Strategies Used on Learning Outcomes” we discuss prior work that has explored the effectiveness of quantity of learning strategies used, as well as the impact of using diverse combinations of learning strategies on various learning outcomes (e.g., comprehension, retention; see lines 155-173). Research questions have also been added to relevant sections.
We revised the paper’s opening paragraph to include our exploration of the influence of the number and diversity of learning strategies used. We believe that these additions increased the coherence of our research questions. We hope you agree.
Comment 3: Do all the previous studies reveal the effectiveness of elaboration strategy? Are there any studies that have not found this strategy to improve test performance? And why? I believe they may be useful to provide an explanation for the results of this study.
Response 3: Several reviews of the learning strategies literature explain why elaboration may not yield positive performance outcomes. In line with your suggestion, in our Future Research Directions, we add another explanation as to why we did not find support for our hypothesis, which can be found below:
“Another reason why we did not find an effect of elaboration prompt on performance may be due to learners requiring more instruction to successfully implement the prompt [16]. This is especially important for learners with low prior knowledge of the content domain because effectiveness of learning strategies depends on quality of elaborations generated by the learner [11]. Future research could explore how varying levels of instruction and subsequent skill at elaboration influences performance.” (see lines 724-730).
Comment 4: The section of Data Analysis is in lack. The authors should elaborate on what statistical methods and software they used to analyze the data.
Response 4: In lines 321-368 (Section 2.5) we have included an additional section titled Data Analysis. In this section, we elaborate on what statistical methods and software we used to test our hypothesis and explore our research questions.
Comment 5: I am confused about the operational definition and the measurement of the Number of Learning Strategies Used. In my view, the number of strategies refers to the frequency of strategies use. Maybe it would be more appropriate to use the Variability of Learning Strategies Used.
Response 5: Thank you for your suggestion to clarify our operationalization and measurement of Number of Learning Strategies Used. We agree that we did not appropriately define this construct, and in the revised version of the manuscript (see lines 547-548), we operationalize it as the number of learning strategies participants reported in their open-ended responses. We also provide additional details in our newly added Data Analysis section detailing how we measured the construct.
However, we disagree with naming Number of Learning Strategies Used as Variability of Learning Strategies Used. Instead, we have renamed Number of Learning Strategies Used to Quantity of Learning Strategies Used. We felt the words variability and diversity were too close in meaning, and may further confuse the reader when interpreting our exploratory analyses.
Comment 6: In 3.1.5, I am concerned that the factor of Variability of Learning Strategies would confound the results when comparing test performance among the Mixed Group, Generative Group, and Non-Generative Group, as the Mixed Group tended to use more types of learning strategies. An ANOVA test should be conducted after controlling the factors of the Variability of Learning Strategies.
Response 6: Thank you for this comment because it made us consider the drivers behind some of our effects. To address this concern, we first examined whether the generative, non-generative, and mixed groups did indeed report using more or fewer learning strategies, and confirmed your suspicion that the mixed group used significantly more (M = 0.42, SD = 0.56 for the non-generative group; M = 1.16, SD = 0.37 for the generative group; M = 2.56, SD = 0.96 for the mixed group; F(2,158) = 102.3, p < .001). We now report these findings on lines 569-572. We further conducted an ANCOVA to examine whether controlling for the quantity of learning strategies affected our results, and indeed, after controlling for quantity of learning strategies, the significant effect of mixed learning strategy use was rendered non-significant, a finding we now report on lines 572-577. We continue to include all of these exploratory analyses in the paper because we do not believe our results provide conclusive evidence that either quantity or variety of learning strategy use is better given that they are confounded in our analysis. We call for future experimental research to tease apart these effects in the Discussion section (lines 654-657).
Round 2
Reviewer 2 Report (Previous Reviewer 2)
Comments and Suggestions for Authors
Thank you for your thorough revision of the manuscript. I appreciate the efforts you have made to address the concerns. I agree the manuscript can be accepted with the following minor revisions:
1. In Abstract, Line 22-23, “the number quantity and diversity of learning strategies used significantly influenced knowledge test performance, and that quantity and diversity of learning strategy use are related”. This statement might not be appropriate, given that the diversity of learning strategy did not significantly influence performance after controlling the quantity.
2. In 1.2, the authors only put forward the hypothesis without the research question. In 1.3 and 1.4, the authors only put forward research questions without corresponding hypotheses. Both research questions and hypotheses should be presented.
Author Response
Thank you for your thorough revision of the manuscript. I appreciate the efforts you have made to address the concerns. I agree the manuscript can be accepted with the following minor revisions:
We appreciate your thorough review of our work. Your feedback has refined our work significantly. Our responses are in red below.
Comment 1: In Abstract, Line 22-23, “the number quantity and diversity of learning strategies used significantly influenced knowledge test performance, and that quantity and diversity of learning strategy use are related”. This statement might not be appropriate, given that the diversity of learning strategy did not significantly influence performance after controlling the quantity.
Response 2: We have revised our abstract based on your feedback. Now, in lines 21-24, we state, "Further exploratory analyses found that the quantity and diversity of learning strategies used individually influenced knowledge test performance. ANCOVA results revealed, however, that when controlling for quantity, the diversity of learning strategies used did not significantly influence knowledge test performance."
Comment 2: In 1.2, the authors only put forward the hypothesis without the research question. In 1.3 and 1.4, the authors only put forward research questions without corresponding hypotheses. Both research questions and hypotheses should be presented.
Response 2: In Section 1.2, we added a sentence that details one research question we made a direct hypothesis about to further test. "Our research question is as follows: Do prompts that require learners to connect to-be-learned material with personal experiences influence learning outcomes, particularly, performance on a knowledge test?" (see lines 129-131).
We refrained from developing and presenting hypotheses in Sections 1.3 and 1.4 because these sections are dedicated to our exploratory analyses. We only presented our research questions in Sections 1.3 and 1.4, given that these analyses were conducted in an effort to further investigate alternative explanations for our null findings (presented in Section 1.2). However, we believe readers would benefit from increased clarity on when we are stating research questions versus hypotheses. To address this need, on lines 153 to 154, we added the phrase, "with the following research question" when setting the stage for research question 1. Similarly, on lines 174 to 175, we revised this sentence to now say, "As such, in the following research question, we explore the impact of the quantity of learning strategies used on knowledge test performance" before presenting research question 2. Finally, before presenting research question 3, we have revised the sentence on lines 187-189 to now state, "Thus, in addition to quantity, our third research question also explored if the diversity of learning strategies reported influenced learning outcomes".
This manuscript is a resubmission of an earlier submission. The following is a list of the peer review reports and author responses from that submission.
Round 1
Reviewer 1 Report
Comments and Suggestions for Authors
Thank you for the opportunity to review the manuscript. I believe the study is solid and has the potential to be published. However, several issues need to be addressed in a major revision.
Major Issues:
- The authors reported a significant result with p = .05. This is not considered significant. Did the authors round the number up? Please provide the correct p-value.
- High-quality criteria are missing, such as pre-registration, ethics approval, and power analysis. Can the authors provide these?
- I had difficulty reading the abstract and introduction due to some incoherence and structural issues:
o The introduction should start with a broad topic and conclude with the research questions and aims of the current study. In this manuscript, the current study is presented very early and in too much detail (e.g., explanation of the I-O, SOI model), which is distracting and confusing at this stage.
o I would expect a structure like which would help me, as a reader, to follow your story: "Generative learning strategies, such as elaborating, can be effective for learning. However, most students do not apply generative learning strategies on their own. Prior research suggests prompting such strategies to enhance students’ learning. In this study, we prompt students to elaborate on new content to investigate the effectiveness of prompting."
o I also struggled to understand the study's aim because the authors were not consistent with some concepts. Terms like elaboration strategy use, self-explaining, generative learning, and cognitive learning strategy were used interchangeably. As a researcher of generative learning activities, I see elaborations as part of generative learning strategies. Generative learning strategies can be applied by students (e.g., generating an explanation), and within this explanation, they may elaborate on the content (or not). Please be clearer with these concepts and revise your manuscript (not only the introduction) accordingly.
- Theoretical Background:
o The authors focused primarily on Fiorella & Mayer. Please include a broader range of research literature on generative learning (e.g., Brod, 2020: Generative Learning: Which Strategies for What Age?) and self-explaining (e.g., McNamara: Self-Explanation and Reading Strategy Training (SERT) Improves Low-Knowledge Students’ Science Course Performance).
o Minor: It would be helpful to start with the generative learning theory and then explain the SOI model.
- Coder Agreement (line 313): The final coder agreement remains at .66? This is quite low. I strongly recommend coding the answers again since all the following analyses are based on this coding.
- Explorative Analyses: I appreciate that the authors made their exploratory analyses transparent. However, the extent of the exploratory analyses (7.5 pages out of 16) is excessive. The authors had their research question and hypothesis, which they tested and answered. The subsequent exploratory analyses go far beyond the stated research questions and are not included in the theoretical background. Exploratory analyses can be interesting, and the use or degree of using strategies may mediate the effect, but I strongly recommend the authors restrict their exploratory analyses to a minimum. In this paper, it appears to be a completely new research question that should be addressed in a different study.
Minor Issues:
- Line 9: I would delete "allowing for multiple retrieval routes when recalling information" as it provides too much detail, which remains unclear in the abstract. At least, I am not sure what the authors mean by this sentence.
- The results of the thematic analyses are missing in the abstract.
- Line 78-80: In what relation is this sentence to self-explaining? It does not seem connected to the content (prompting self-explanation) but rather refers to the broader picture of generative learning.
- Line 81-83: Why is elaborating more effortful for learners compared to drawing?
- Line 103-105: This sentence is a general statement about “self-explaining” but not “prompting self-explaining.” It feels like topics got mixed up. There is research on self-explaining (and its effectiveness) and then on additional prompts during self-explaining. Maybe deleting the word “prompt” in the manuscript would solve this confusion. Alternatively, both terms should be used with more precision.
- Line 113-125: First, the authors state that identifying differences between concepts is more effective than comparing concepts, but then the students need to find differences and agreements. Why both when identifying differences is more effective?
- Line 141: Gender is not an interval-scaled variable and should be treated like the variable race/ethnicity.
- Figure 1: 10 minutes in the figure for the study phase versus 8 minutes in the text of the manuscript.
- 2.3 Measures a) Prior I-O Psychology Knowledge: Does the final test contain 16 items in total? Did you measure a sum score or a mean score? Please indicate the maximum score of the test and provide more details.
- 2.3 Measures b) Final Knowledge Test Performance: Does this test contain 23 items in total? Is the maximum score 23? Did students get 1 point for a correct answer? Please provide more details.
- What result do you get when using McDonald’s Omega? This reliability test is also appropriate for dichotomous items.
- Line 222-224: What questions did the students get?
- Table 1: As a reader, it is helpful to find the scale (e.g., 0-5) after the variable within a table to judge the data more appropriately.
- Table 2: What do the values in brackets represent? This cannot be a correlation as an item cannot correlate with itself.
Comments on the Quality of English LanguageOnly minor issues (sometimes present instead of past; once a word was missing).
Reviewer 2 Report
Comments and Suggestions for Authors
Thanks for the opportunity to read this paper. It reported how the prompt of using the elaboration strategies potentially improved knowledge test performance. It also explored the additional strategies students used when reading the passage. This study has the potential to contribute to our understanding of the relationship between elaboration strategies use and performance, as well as the types of strategies students may use when completing tasks. Despite these strengths, some revisions are needed. I have several concerns which will be described in detail in the following:
1. The authors reviewed the relationship between Fiorella and Mayer's eight learning strategies and learning outcomes. However, it is essential to specify which domains these outcomes pertain to. Are there any studies within the I-O psychology domain? Additionally, what age groups were these studies focused on?
Moreover, the current research mainly focuses on the elaboration strategy. It is important to discuss the unique significance or importance of the elaboration strategy in I-O psychology learning.
2. This study uses prompts to create examples that connect to personal lives. In the Introduction, only differential-associative prompts and example elaboration prompts are mentioned, with a note that differential-associative prompts are more effective. Does the prompt used in this study belong to differential associative prompts or example elaboration prompts? If neither, have previous studies employed this type of prompt, and how is this type of prompt related to learning outcomes?
3. Five additional participants were removed due to missing responses on the final knowledge test. What was the extent of their missing responses? Were they absent the entire test or only parts of it? Additionally, were these participants homogeneous with the others who had complete data in terms of prior knowledge?
4. Further details about the participants are needed. From which department were these participants recruited? Did the sample include students who major in psychology?
5. The intervention was conducted only once and lasted for just 10 minutes. This might be a reason for the insignificant difference in test performance between the intervention and control groups. This limitation should be discussed.
6. The materials used for the intervention need to be described in the methods section. What was the content of these materials, and how did they relate to the participants' prior knowledge and the test? How did students employ the elaboration strategy with these materials? More details are needed.
7. While the intervention group reported more elaboration strategies used immediately after the intervention, the key issues are the sustainability and transferability of these strategies. This study does not address these aspects, and this limitation should be more discussed.
8. Instead of zero-order correlations among prior knowledge, test performance, and elaboration strategy use, consider conducting a partial correlation analysis. This would control for prior knowledge and provide a more accurate relationship between test performance and elaboration strategy use.
9. In thematic analysis, the dimension of 'connection' appears similar to 'elaboration.' What are the distinguishes between the two?
10. The study categorizes the repetition strategy as Ineffective Strategy Use. However, repetition can be effective in some fundamental tasks. Again, it ties into the nature of the intervention materials. What are the characteristics of these materials?
11. Besides theoretical implications, the practical significance of the findings should be discussed.
12. A thorough discussion is needed on why there were no significant differences between the intervention and control groups, as well as the positive relationship between strategy use and test performance. Consider the specific characteristics of the intervention group and domain compared to previous research, and the potential limitations of the experimental design (e.g., limited rounds of intervention, lack of assessment of strategy use sustainability). These factors might explain why the intervention group's use of elaborate strategies increased, but there was no significant improvement in their test performance in the short term.
Reviewer 3 Report
Comments and Suggestions for Authors
The work is very valuable and original. Well exposed and with tables and figures that enrich the work. I would only highlight as an improvement the conclusions, which for me are scarce. Taking into account the results of the research.
Round 2
Reviewer 1 Report
Comments and Suggestions for Authors
The authors carefully revised their manuscript, which improved its quality.
Author Response
Comment 1: The authors carefully revised their manuscript, which improved its quality.
Response 1: Thank you again for taking the time to review our work. We appreciate the constructive feedback and agree that the revisions significantly improved the manuscript quality.
Reviewer 2 Report
Comments and Suggestions for Authors
Thank you for your thorough revision of the manuscript. I appreciate the efforts you have made to address the concerns raised in the previous version of the manuscript. However, I have identified two minor issues that need to be revised.
1. Under section 2.2 "Materials," there is only one subheading, 2.2.1 "Passage Information." It may be better to remove the subheading 2.2.1 and merge its content directly under 2.2.
2. In Table 1, there is an inconsistency in the way numerical values are presented; some numbers include a leading zero before the decimal point while others do not. For clarity and aesthetic consistency, it would be beneficial to standardize the formatting of these values.
Author Response
Thank you for your thorough revision of the manuscript. I appreciate the efforts you have made to address the concerns raised in the previous version of the manuscript. However, I have identified two minor issues that need to be revised.
Comment 1: Under section 2.2 "Materials," there is only one subheading, 2.2.1 "Passage Information." It may be better to remove the subheading 2.2.1 and merge its content directly under 2.2.
Response 1: Thank you for this suggestion. We have removed the subheading 2.2.1 "Passage Information" and merged it with content directly under heading 2.2 "Materials" (see lines 747-765).
Comment 2: In Table 1, there is an inconsistency in the way numerical values are presented; some numbers include a leading zero before the decimal point while others do not. For clarity and aesthetic consistency, it would be beneficial to standardize the formatting of these values.
Response 2: We agree there were inconsistencies in how numerical values were presented in Table 1, particularly in our reporting of Cohen's d. We have added a leading zero for each effect size, means, and standard deviations. We did not add a leading zero in the column where we report p-values due to APA reporting standards (p-values do not exceed 1, so they do not need a leading zero). Thank you for drawing our attention to this section of our manuscript (see lines 1003-1004).